# Fabrication of Lead-Free Bismuth Based Electroceramic Compositions for High-Energy Storage Density Applications in Electroceramic Capacitors

**Azam Khan** [1,†], **Taj Malook Khan** [2,3,†], **Jianbo Wu** [2,3], **Hazrat Bilal** [1], **Shahan Zeb Khan** [4], **Abdul Manan** [5], **Xiujian Wang** [1,*] **and Noor Shad Gul** [2,3,*]

1    State Key Laboratory for Chemistry and Chemical Engineering of Medicinal Resources, School of Chemistry and Pharmacy, Guangxi Normal University, Guilin 541004, China; 201725058@stu.gxnu.edu.cn (A.K.); hbilal4667@gmail.com (H.B.)

2    Drug Discovery Research Center, Southwest Medical University, Luzhou 646000, China; tajmalook83@swmu.edu.cn (T.M.K.); jbwucn1996@yahoo.com (J.W.)

3    Laboratory for Cardiovascular Pharmacology, Department of Pharmacology, The School of Pharmacy, Southwest Medical University, Luzhou 646000, China

4    Department of Chemistry, University of Science and Technology, Bannu 28100, Pakistan; dr.shahanzeb@ustb.edu.pk

5    Advanced Materials Research Laboratory, Department of Physics, University of Science and Technology, Bannu 28100, Pakistan; drmanan82@ustb.edu.pk

\*    Correspondence: wang1_xj@yahoo.com.cn or wang1_xj@aliyun.com (X.W.); noorshad90@swmu.edu.cn (N.S.G.)

†    These authors contributed equally to this work.

**Abstract:** Lead-based electro-ceramic compositions are excellent energy storage materials used for high-energy storage density applications in dielectric ceramic capacitors. However, these materials have lead contents in their compositions, making them toxic, with a negative impact on human health and the environment. For this reason, we synthesized a lead-free bismuth-based electro-ceramic perovskite, $0.80(0.92Bi_{1/5}Na_{1/5}TiO_3\text{-}0.08BaTiO_3)\text{-}0.20(Na_{0.73}Bi_{0.09}NbO_{3-x}Ta_2O_5)$, abbreviated (BNT-BT-NBN$_{1-x}$T$_x$), from mixed oxides with doping of tantalum (Ta) at different concentrations, using a conventional solid-state reaction method. The effects of Ta doping on the phase evolution, microstructure development, and energy storage applications were investigated. Detailed powder X-ray diffraction analysis revealed a pure perovskite phase with Ta doping at ≤0.05. Furthermore, it was observed that excessive addition of Ta has been resulted in secondary phase generation. Scanning electron microscopy validated the development of dense microstructures with a reduced grain size for the Ta concentration of ≤0.01. Electrochemical analysis revealed a maximum polarization ($P_m$) of ~22 μC/cm² and a recoverable energy density of 1.57 J/cm³ with 80% efficiency for Ta doping at 0.05 with an applied field of 175 kV/cm. These results demonstrate the development of enhanced ferroelectric characteristics in an as-synthesized electro-ceramic perovskite for high-energy storage density applications in electro-ceramic capacitors.

**Keywords:** electro-ceramic composites; energy storage density; discharging power density; dielectric ceramic capacitor

## 1. Introduction

Currently, 11% of consumable energy is provided by renewable resources, while the remaining 84% is still obtained via the burning of fossil fuels such as oil, coal, and gas. Consequently, the burning of fossil fuels for energy purposes also discharges an enormous amount of carbon dioxide, which is considered the most harmful gas among the greenhouse gases and accountable for global warming and climate change [1]. Consequently, it is essential to reduce temperature to less than 2 °C, as signed and agreed by world leaders at

the Paris Agreement, with the aim of decreasing carbon dioxide emissions by 45% by 2030 and reaching net-zero emissions by 2050 [2,3]. In this perspective, advances in renewable-harvesting technologies can play a significant role in achieving the objectives of the Paris Agreement, by obtaining greater amounts of clean energy from renewable resources such as the sun, wind, and tide. Therefore, in order to reduce the burning of fossil fuels and harvest renewables for clean energy, it is of utmost importance to integrate energy-harvesting technologies with energy storage devices, particularly for the electrification of power trains, planes, ships, electric cars, and military vehicles for shipment and transportation purposes. For this purpose, harvesting renewables is crucial for lessening the dependency on fossil fuel usage, which will ultimately result in carbon dioxide settling in the upper atmosphere, a key step towards the remediation of global warming and climate change [4–6].

Currently, various composites and condensed solid solutions, such as polymeric and ceramic compositions, are used in high-energy storage devices for high-energy storage density applications [7]. In this regard, lead-based electro-ceramics have exhibited excellent energy storage density characteristics and applications; however, lead-based electro-ceramic materials have toxicity due to their lead contents, posing human health risks in consumer-level devices such as capacitors and actuators. Despite having excellent properties, lead-based electro-ceramics pose human health risks and environmental concerns [8]. Therefore, lead-free alternatives having the characteristics of nontoxicity, eco-friendliness, and high-energy storage density applications are sought [9]. In this perspective, bismuth-based electro-ceramic compositions have recently received significant research interest. Consequently, a tremendous amount of current research has focused on bismuth-based electro-ceramic compositions owing to their excellent, eco-friendly ferroelectric and piezoelectric properties for high-energy storage density applications in energy storage devices [10].

Currently, various energy storage devices such as fuel cells, batteries, and capacitors are applied for high-energy storage density applications [11]. Among them, dielectric ceramic capacitors have demonstrated outstanding features of a high power density, good exhaustion and fatigue resistance, fast charging/discharging capacity, and longer life consistency [12]. Dielectric capacitors are typically manufactured either from polymeric or condensed ceramic materials [13]. Polymeric-based capacitors have a high output power at room temperature; however, they cannot withstand high-temperature (~100 °C) conditions [14]. Nonetheless, ceramic-based capacitors are equally workable under high-temperature conditions [15].

The storage of electric energy per unit volume by a polarizing dielectric material with an external electric field is known as the energy storage density ($W_s$), and the amount of energy released per unit volume via a depolarizing dielectric by switching off an external applied electric field is recognized as the recoverable energy density ($W_{rec}$). Both energy densities, $W_s$ and $W_{rec}$, are determined from the polarization vs. electric field (P–E) hysteresis loop using Equations (1) and (2) [16,17], as follows:

$$W_s = \int_0^{P_{max}} E dP \tag{1}$$

$$W_{rec} = \int_{P_r}^{P_{max}} E dP \tag{2}$$

where $P_r$ represents remnant polarization, also known as residual polarization; $P_{max}$ stands for maximum polarization, also known as saturation polarization; and E represents the external applied electric field. The efficiency of dielectric ceramics is determined by Equation (3), as follows:

$$\eta = \frac{W_{rec}}{W_s} \times 100 \tag{3}$$

It is obvious from Equation (2) that ceramic materials with zero residual polarization are preferable for realizing a high recoverable energy density. Further, dielectric ceramic

materials that are almost linear satisfy this phenomenon; however, linear dielectric ceramics have a lower permittivity ($\varepsilon_r$), due to which their recoverable energy density is limited. Although linear dielectric ceramic materials exhibit a high dielectric breakdown strength (DBS) and efficiency ($\eta$), their $P_{max}$ is still small; hence, they are not feasible for high-recoverable-energy storage density applications [18]. It is obvious from Equation (1) that $W_s$ depends both on a high $P_{max}$ and the applied electric field E; nonetheless, $W_{rec}$ depends on the maximal value of $\Delta P = P_{max} - P_r$, which is only achievable by remanence-free ceramic materials. Consequently, it is crucial to optimize both of the above parameters in order to obtain a high energy storage density and recoverable energy density in dielectric ceramic capacitors. Based on their polarization vs. electric field (P–E) hysteresis loops, dielectric ceramic materials are classified into two main categories: linear and nonlinear dielectrics [19,20]. The polarizations and electric fields are directly proportional in linear dielectric ceramics, though linear dielectrics have a high DBS; however, they have a lower dielectric permittivity ($\varepsilon_r$) and $P_{max}$, hence performing poorly in energy storage applications [21]. In the case of nonlinear dielectric ceramic materials, the polarization and applied electric field are independent of each other [22]. For the purpose of carrying out steady and comprehensive research, nonlinear dielectric ceramics are further divided into four types: paraelectrics, ferroelectrics (FEs), relaxor ferroelectrics (RFEs), and anti-ferroelectrics (AFEs) [16,23]. At the macroscopic level, both paraelectric and ferroelectric materials are alike; however, at the microscopic level, spontaneous polarization is absent in paraelectric materials, and the net polarization becomes zero as soon as the external applied electric field is removed. Moreover, ferroelectric materials retain their polarization even after the removal of the external electric field. This is the reason why paraelectric ceramics are not feasible for energy storage applications. A large number of ferroelectric ceramics have a large $P_r$ and small DBS, due to which their energy storage density is low [24]. The double P–E hysteresis loop permits anti-ferroelectric (AFE) materials to store a high power density, but the majority of these materials have lead contents in their compositions that can cause toxicity and pose human health risks and severe environmental impacts [25]. In the case of lead-free AFEs, the phase transition of anti-ferroelectrics to ferroelectrics is a more challenging phenomenon due to the long-term cycling operations. Therefore, only relaxor ferroelectric materials have the characteristics of highly polar nano-regions (PNRs), minimal $P_r$, small $E_c$, and high DBS, making them feasible for high-power storage density applications. As a consequence, an increasing amount of research interest has been directed towards lead-free relaxor ferroelectric ceramics recently for high-energy storage density and recoverable energy density applications [26].

Currently, the lead-free bismuth-based ceramic composition ($Na_{0.5}Bi_{0.5}$) $TiO_3$ (NBT) has been studied comprehensively owing to its excellent ferroelectric characteristics such as its maximal polarization ($P_{max}$~40 $\mu C/cm^2$) and high Curie temperature ($T_c$~320 °C); however, it has high field coercivity ($E_c$) and poor piezoelectric properties. Hence, it has been recommended to dope it with other perovskite solid solutions for high-energy storage density and recoverable energy density applications [27,28]. To introduce relaxor behaviors in NBT, it has been modified via doping with other perovskite materials such as $Bi_{0.5}K_{0.5}TiO_3$ (BKT), $K_{0.5}Na_{0.5}NbO_3$ (KNN), and $BaTiO_3$ (BT) [29,30]. In this regard, Praharaj et al. [31] prepared $0.76Na_{0.5}Bi_{0.5}TiO_3$-$0.2SrTiO_3$-$0.04BaTiO_3$ by doping $BaTiO_3$ and $SrTiO_3$ into the NBT structure, which resulted in the development of a morphotropic phase boundary (MPB). Ceramic compositions with an MPB are typically applied as matrices in high-energy storage density applications owing to their pinched (P–E) loops, high $P_{max}$, and minimal $P_r$ [32,33]. In this connection, Li et al. [34] fabricated a lead-free ternary ceramic composition (1 − x) (BNBT6-xCZ) by doping $CaZrO_3$ in a binary ceramic system, which demonstrated an energy storage density of $W_s$~0.70 $J/cm^3$ at a low electric field of 70 kV/cm. Likewise, Wang et al. [35] reported a ternary ceramic system of (1 − x) $Bi_{0.47}Na_{0.4}7Ba0.06TiO_3$-$KNbO_3$ (BNBT-xKN, x = 0–0.08) with $W_s$~0.89 $J/cm^3$ at an applied electric field of 100 KV/cm, by adding $KNbO_3$ in a binary ceramic system of BNT-BT. In another similar study, Zhu et al. [36] synthesized a lead-free NBT-based

ternary ceramic system, BNT-ST-BMH, by adding $SrTiO_3$ and $Bi(Mg_{0.5}Hf_{0.5})O_3$, which displayed a high $W_{rec}$ of ~5.59 $J/cm^3$ with 85.3% efficiency. Likewise, Zhang et al. [37] studied the ternary ceramic system $(1 - x)$ (0.775NBT-0.225BSN)-xBZ by adding $BaZrO_3$, which achieved a high Ws of ~2.08 $J/cm^3$ and 88.8% efficiency at an applied electric field of 245 kV/cm. In another study, Nilkhao et al. [38] reported the partial doping of Ta in Nb in $0.9(0.92Bi_{0.5}Na_{0.5}TiO_3$-$0.08BaTiO_3)$-$0.01(NaNb_{1-3x}Ta_xO_3)$, which achieved an energy storage density of $W_s$~0.37 $J/cm^3$ and efficiency of 75.1%. In the same way, Li. et al. [39] fabricated a binary ceramic system with doping of La, $(1 - x)$ $(0.72Bi_{0.5}Na_{0.5}TiO_3$-$0.28$ $Bi_{0.2}Sr_{0.7-0.1}TiO_3)$-xLa, which achieved an energy storage density of 1.2 $J/cm^3$ at an applied field of 90 kV/cm. Similarly, Khan et al. [40] fabricated an NBT-based ternary ceramic composition, $(1 - z)$ $(0.94Na_{0.5}Bi_{0.5}TiO_3$-$0.06BaTiO_3)$-$zNd_{0.33}NbO_3$, which achieved an excellent recoverable energy density of 2.01 $J/cm^3$ and efficiency of 62.5% with a z content of 0.02 at an applied field of 250 kV/cm. Likewise, Zhang et al. [41] fabricated a lead-free binary ceramic composition, $(1 - x)$ $Na_{0.5}Bi_{0.5}TiO_3$-$xBaHfO_3$, abbreviated as $(1 - x)$ NBT-xBH, which showed a recoverable energy density of 2.1 $J/cm^3$ at a low applied field of 175 kV/cm. Meanwhile, using the same conventional solid-state reaction method, Wang et al. [42] synthesized a binary ceramic system by doping $CaTiO_3$ at a level of 0.2, $[0.8Bi_{0.5}Na_{0.5}TiO_3$-$0.2CaTiO_3$, $(0.8BNT$-$0.2CT)]$, which achieved an energy storage density of $W_s$~1.38 $J/cm^3$ with an excellent efficiency of 91% at a low applied field.

Here, we fabricated an electro-ceramic composition, $0.80(0.92Bi_{0.5}Na_{0.5}TiO_3$-$0.08BaTiO_3)$-$0.20Na_{0.73}Bi_{0.09}NbO_3$, abbreviated as (NBT-BT-NBN), using a conventional solid-state reaction method, by doping Ta in (BNT-BT-$NBN_{1-x}T_x$) in a series of x = 0.00, 0.05, 0.10, 0.15, 0.20, in order to investigate the doping effects of Ta on the as-synthesized ceramic compositions' phase evolution, microstructure development, dielectric characteristics, and energy storage applications in electro-ceramic capacitors [43].

## 2. Results and Discussions

### 2.1. Phase Examination

Figure 1 depicts the PXRD patterns of the ceramic compositions of (NBT-BT-$NBN_{1-x}T_x$) in a series of (x = 0.00, 0.05, 0.10, 0.15, 0.20) recorded at room temperature with 2θ in the range of 10–70°.

The X-ray pattern of each ceramic composition matched with that of PDF Card No. 070-4760, which was attributed to the single-perovskite structure of $Na_{0.5}Bi_{0.5}TiO_3$ that appeared in the tetragonal phase; however, its peaks shifted towards lower values of 2θ. The main reason for the shifts in these peaks is the substitution of bigger cations of $Ba^{2+}$ (1.61 Å) for $Bi^{3+}$ (1.40 Å), and those of $Na^{1+}$ (1.60 Å) and $Nb^{5+}$ (0.64 Å) for $Ti^{4+}$ (0.60 Å) [44]. Figure 1 shows a single perovskite phase that developed as the major phase of $Na_{0.5}Bi_{0.5}TiO_3$, represented by the peaks marked as "*". However, there were some very low intensity secondary peaks marked as "#" that matched with those of PDF Card No. 32-0118, indicating that $Bi_2Ti_2O_7$ bismuth-based pyrochlore also appeared due to the high concentration of $NaBiNbO_3$. The formation of these secondary phases was also reported by Ni et al. [45] while investigating the ceramic composition of $Bi_{(0.5+x)}$ $(Na_{0.82}K_{0.18})_{0.5-3x}TiO_3$. This secondary bismuth-based pyrochlore phase $Bi_2Ti_2O_7$ might develop due to the vaporization of Bi and Na at rising temperatures. Furthermore, the intensity of $Bi_2Ti_2O_7$ was increased with the increasing z content to 0.08 in the current formulation. The second reason for the appearance of secondary phases may be due to the excessive addition of Ta as a dopant, which did not dissolve in the ternary system of x ≥ 0.10. The few strong peaks of secondary phases that appeared at x = 0.20 were possibly due to the excess Ta doping as compared to x = 0.05, where no such secondary phase developments appeared in its XRD pattern.

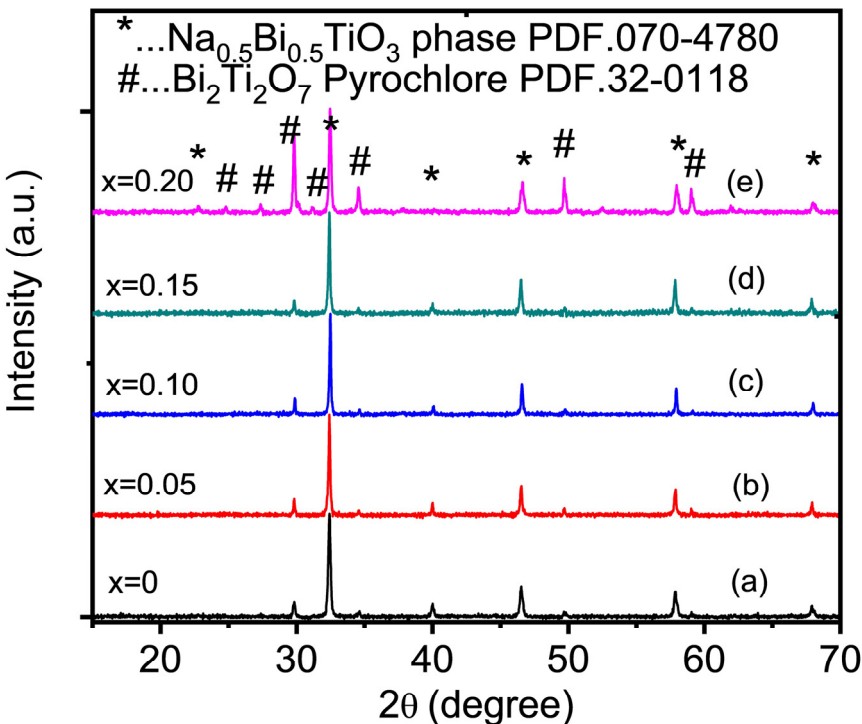

**Figure 1.** PXRD patterns of (NBT-BT-NBN$_{1-x}$T$_x$) ceramic compositions: (**a**) x = 0.00; (**b**) x = 0.05; (**c**) x = 0.10; (**d**) x = 0.15; (**e**) x = 0.20.

*2.2. Microstructure Examination*

Figure 2 shows SEM micrographs of the thermally etched surfaces of all dense ceramic compositions of (NBT-BT-NBN$_{1-x}$T$_x$) in a series of (x = 0.00, 0.05, 0.10, 0.15, 0.20) recorded with a scanning electron microscope. It is obvious from the SEM micrographs that all sintered samples of the ceramic compositions of (NBT-BT-NBN$_{1-x}$T$_x$) had compact, dense microstructures with well-grown grains. The average grain size of each sample was determined via a line intercept method based on SEM photographs at a lower magnification. The mean grain size gradually increased from 1.4 to 1.7 μm for x = 0.00–0.20, respectively. The minimal growth in grain size for x > 0.05 shows that the Ta substitution had little influence on the sintering behavior and grain growth on the microstructures of the as-fabricated ceramic compositions [46]. Bi and Na elements become volatile at high temperatures and induce oxygen vacancies, performing a significant role in the transportation of mass during the sintering process. In the sintering phenomenon, sodium vacancies $V_{Na}$ are formed in the crystal lattice through the volatilization of Na elements. There are defect dipoles formed by oxygen vacancies $V_{Na}$-$V_O$-$V_{Na}$ that compensate the electrical neutrality, which result in small grains due to the grain boundary pinning effect [47,48]. During the coupling of NaBiNbO$_3$ with the BNT-BT system, Nb$^{+5}$ can replace Ti$^{+4}$ to form Nb$_{Ti}$, which neutralizes a fraction of the negative charge $V_{Na}$, reducing the concentration of defect dipoles $V_{Na}$-$V_O$-$V_{Na}$ that result in continuous grain size growth [48,49]. Secondary phases of the bismuth pyrochlore Bi$_2$Ti$_2$O$_7$ that developed in the microstructure were also confirmed via energy-dispersive spectroscopy (EDS), as depicted in Figure 2e.

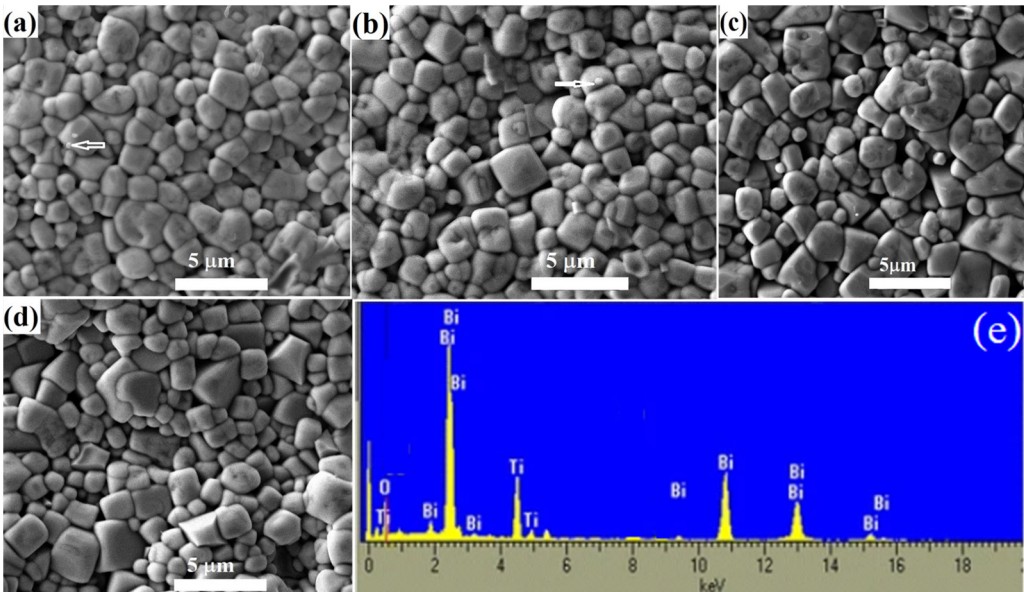

**Figure 2.** Scanning electron micrographs of (NBT-BT-NBN$_{1-x}$T$_x$) ceramic compositions: (**a**) x = 0.00; (**b**) x = 0.05; (**c**) x = 0.10; (**d**) x = 0.15. (**e**) EDS spectrum of the bismuth pyrochlore Bi$_2$Ti$_2$O$_7$ that developed as a secondary phase, indicated by arrows in the microstructures.

*2.3. Dielectric Properties*

Figure 3 shows the dielectric response including the permittivity and dielectric loss for the as-synthesized ternary (NBT-BT-NBN$_{1-x}$T$_x$) (x = 0.00, 0.05, 0.10, 0.15, 0.20) electroceramic compositions. The dielectric permittivity was measured at different frequencies, namely, 1 kHz, 10 kHz, 100 kHz, and 1 MHz, in the temperature range of 20–500 °C. Two separate dielectric peaks in the curves of the dielectric response of BNT-BT ceramic samples have already been reported by Xu et al. [48]. One peak is observed at a lower temperature, denoted as T$_s$, and the other is observed at a high temperature, denoted as T$_m$. Meanwhile, after adding 20% NaNbO$_3$ to BNT-BT, T$_m$ was found to shift to a lower temperature below 50 °C.

All the above ceramic compositions exhibited significant variations in the dielectric permittivity and loss tangent (tan δ) with increasing temperature, which could be attributed to the changes in the structure as well as the local ferroelectric order. A decrease in the dielectric permittivity was observed with increasing x content in the range of 0.00–0.15, along with an increase in the dielectric permittivity with a further increase in the x content to 0.2. The lower content of the Ta doping has resulted in a decrease in the dielectric polarizability per unit volume of Ta [38]. The increase in the relative permittivity ($\varepsilon_r$) with a high content of the Ta dopant may be due to the formation of secondary phases, as depicted in the XRD patterns. The Ta doping shifted T$_m$ from 45 °C to 80 °C. Here, T$_m$ showed broad permittivity peaks and frequency dispersion. However, at an x content of 0.20, the observed $\varepsilon_r$ was higher than that of x < 0.2 compositions. Moreover, there was a decreasing trend in $\varepsilon_r$ for each ceramic sample with increasing frequency. This could be ceased at certain polarizations with higher frequencies. The dielectric loss also deceased with Ta doping at x ≤ 0.05, as shown in Figure 4a,b, but further increases in Ta doping had little effect on the dielectric loss. The tan δ loss was the lowest beyond x ≥ 0.10, as depicted in Figure 4c–e.

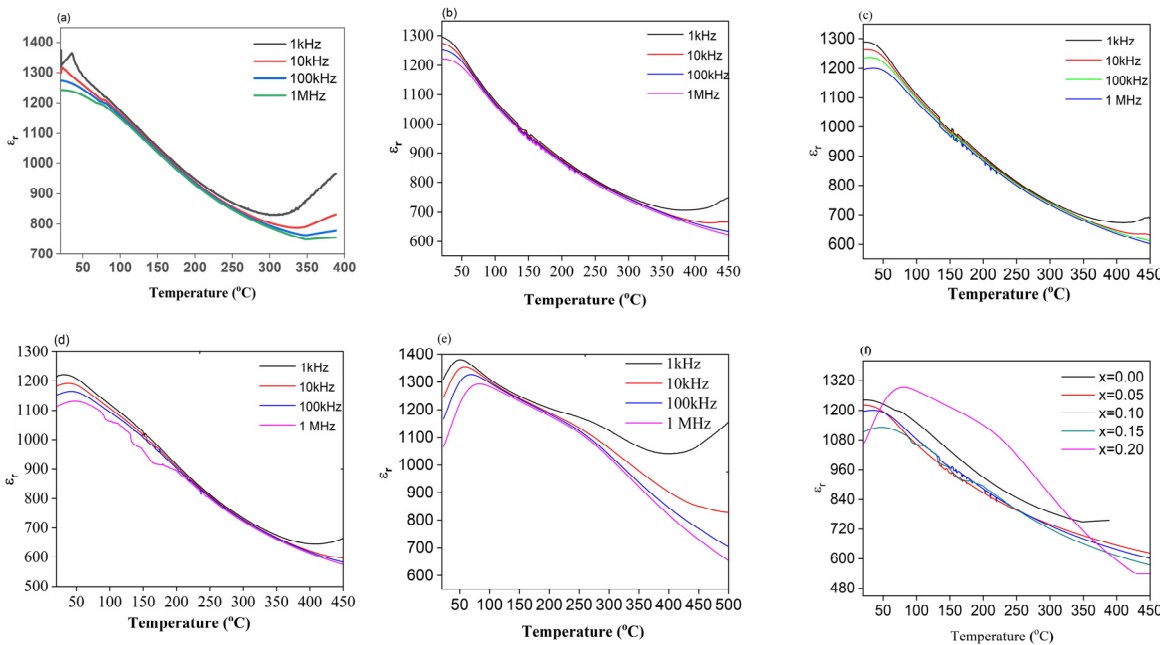

**Figure 3.** Changes in $\varepsilon_r$ for (NBT-BT-NBN$_{1-x}$T$_x$) ceramic compositions: (**a**) x = 0.00, (**b**) x = 0.05, (**c**) x = 0.10, (**d**) x = 0.15, and (**e**) x = 0.20, as a function of the temperature at different frequencies. (**f**) Variations in $\varepsilon_r$ for (NBT-BT-NBN$_{1-x}$T$_x$) ceramic compositions (x = 0.00, 0.05, 0.10, 0.15, 0.2) vs. temperature at 1 MHz.

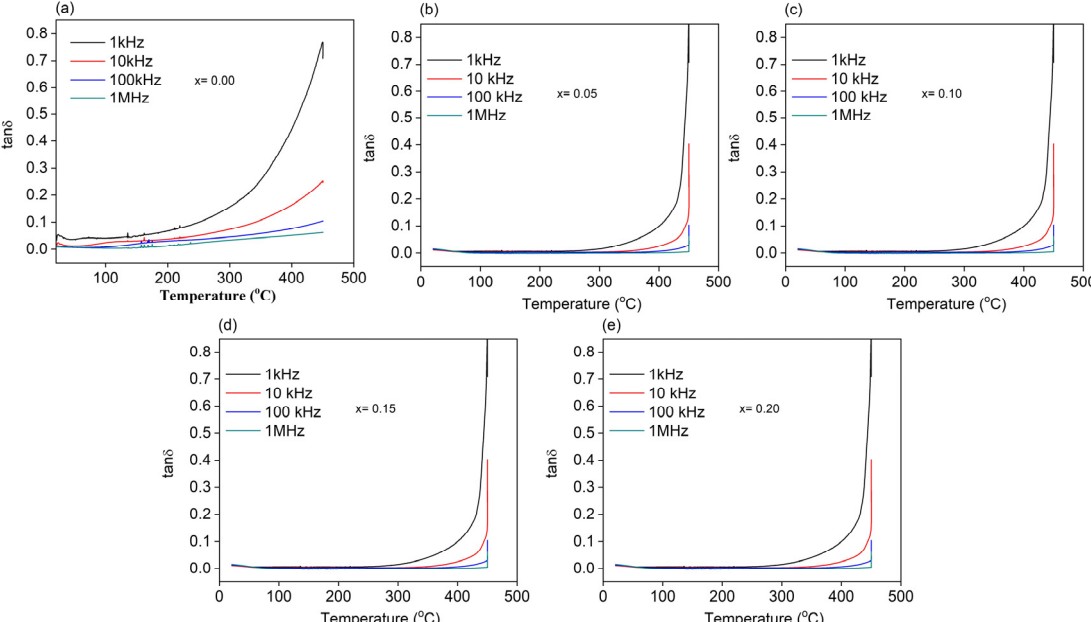

**Figure 4.** Variations in tan δ for (NBT-BT-NBN$_{1-x}$T$_x$) ceramic compositions: (**a**) x = 0.00, (**b**) x = 0.05, (**c**) x = 0.10, (**d**) x = 0.15, and (**e**) x = 0.2, as a function of the temperature with increasing frequencies.

### 2.4. P–E Hysteresis Loops and Energy Storage

The polarization vs. electric field (P–E) hysteresis loops for the (NBT-BT-NBN$_{1-x}$T$_x$) (x = 0.00, 0.05, 0.10, 0.15, 0.20) ceramic compositions are depicted in Figure 5a. The P–E loops were measured at a frequency of 10 Hz for each ceramic composition with a 0.3 mm thickness. Large variations in the maximum polarization (P$_m$) and breakdown electric fields (E$_b$) were observed for each ceramic composition. The energy storage characteristics, such as energy storage/charged density (W$_s$), recoverable or discharge energy density

($W_{rec}$), and efficiency ($\eta$), of each ceramic sample were determined from its P–E hysteresis loop area. Each sample has its own breakdown voltage related to the average grain size and density of the sample. The P–E loop for each composition was measured at its breakdown voltage.

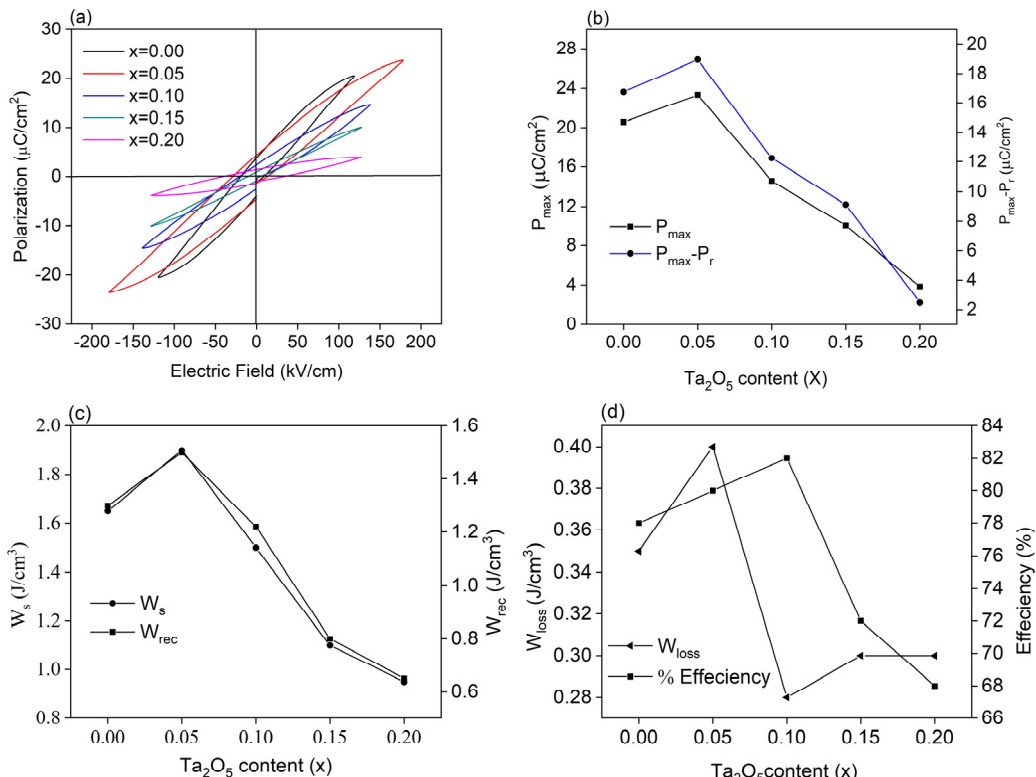

**Figure 5.** (**a**) Variations of polarization for a series of (NBT-BT-NBN$_{1-x}$T$_x$) (x = 0.00, 0.05, 0.10, 0.15, 0.20) ceramics composition as a function of the electric field. (**b**) Variations in $P_{max}$ and $P_{max}$-$P_r$ for (NBT-BT-NBN$_{1-x}$T$_x$) with an increasing x content. (**c**) $W_s$ and $W_{rec}$ with an increasing x content. (**d**) Changes in $W_{loss}$ and $\eta$ for (NBT-BT-NBN$_{1-x}$T$_x$) with an increasing x content.

Figure 5b shows that the maximum polarization for x = 0.00 was 20 $\mu$C/cm$^2$. Meanwhile, there was somewhat of an increase in $P_{max}$ to 22 $\mu$C/cm$^2$ for x = 0.05 at a breakdown field of 170 kV/cm, which then decreased after further increasing the x content to 0.20. The value of $P_{max}$-$P_r$ was also higher for x = 0.05, and these high values of $P_{max}$ and $P_{max}$-$P_r$ for x = 0.05 contributed to high charge power and recoverable energy densities at a low external electric field of 170 kV/cm.

## 2.5. Energy Storage Density Analysis

The energy storage density ($W_s$) and recoverable energy density ($W_{rec}$) were calculated by integrating the P–E hysteresis loop areas of each ceramic composition. Figure 5c displays the variation in $W_s$ and $W_{rec}$ for the (NBT-BT-NBN$_{1-x}$T$_x$) (x = 0.00, 0.05, 0.10, 0.15, 0.20) ceramic compositions as a function of the x content. In this work, $W_s$~1.96 J/cm$^3$, $W_{rec}$~1.57 J/cm$^3$, and an efficiency ($\eta$) of 80% were obtained for x = 0.05. The large values of $W_{rec}$ and $W_s$ for x = 0.05 could be credited to the maximal values of $P_{max}$ and $P_m$-$P_r$ as compared to other compositions with higher values of x in this research work. The variation in the energy density loss ($W_{loss}$) and efficiency ($\eta$) for the (NBT-BT-NBN$_{1-x}$T$_x$) (x = 0.00, 0.05, 0.10, 0.15, 0.20) ceramic compositions as a function of the x content is shown in Figure 5d. In this research work, the maximum efficiency of 80% was achieved by the ceramic composition x = 0.05.

A comparative analysis of the current study with previously published works on NBT-based ceramics is presented in Table 1. The analysis shows that the $W_{rec}$ obtained for

(NBT-BT-NBN1-xTx) (x = 0.05) in the present study is higher than that described for other NBT-based ceramic systems.

**Table 1.** The comparative analysis of the current study with previously published works on NBT-based ceramics.

| Materials | Dielectric Breakdown Field (kV/cm) | $W_{rec}$ (J/cm$^3$) | Ref. |
|---|---|---|---|
| 0.95[0.94BNT-0.06BT]-0.05LT | 107 | 1.65 | [49] |
| 0.95[BNT-0.06BT]-0.05AN | 72 | 0.71 | [50] |
| 0.99(0.7NBT-0.3Bi$_{0.2}$Sr$_{0.7}$TiO$_3$)-0.01NN | 88 | 1.04 | [51] |
| 0.94(0.75BNT-0.25BKT)-0.06BA | 106 | 1.15 | [52] |
| 0.89BNT-0.06BT-0.05KNN | 55 | 0.60 | [53] |
| 0.837BNT-0.063BT-0.1KNN | 52 | 0.45 | [54] |
| 0.94[0.93NBT-0.07BT]-0.04LZ | 99 | 1.19 | [55] |
| Bi$_{0.4806-x}$Yb$_{0.05}$Na$_{0.4606}$Ba$_{0.0588}$Ti$_{0.98}$Al$_{0.02}$O$_3$ | 96 | 0.70 | [56] |
| 0.80(0.92Bi$_{0.5}$Na$_{0.5}$TiO$_3$-0.08BaTiO$_3$)-0.20 Na$_{0.73}$Bi$_{0.09}$Nb$_{0.95}$Ta$_{0.05}$O$_3$ | 175 | 1.57 | this work |

### 3. Experimental Procedures

#### 3.1. Materials

The chemicals and reagents used in this research work were sodium carbonate, $Na_2CO_3$ (99.9%), barium carbonate, $BaCO_3$ (99.5%), bismuth oxide, $Bi_2O_3$ (99%), titanium dioxide, $TiO_2$ (99.8%), niobium oxide, $Nb_2O_5$ (99%), tantalum oxide, $Ta_2O_5$ (99.8%), ethanol, $C_2H_5OH$, (99.7%), and poly-vinyl alcohol. All these chemicals and reagents were purchased from a Sigma Aldrich manufacturer. Anhydrous ethanol and de-ionized water were used for dissolution purposes.

#### 3.2. Fabrication of [0.80(0.92Bi$_{1/5}$Na$_{1/5}$TiO$_3$-0.08BaTiO$_3$)-0.20(Na$_{0.73}$Bi$_{0.09}$NbO$_{3-}$xTa$_2$O$_5$)]

Several processes are applied to fabricate lead-free electro-ceramic compositions, such as the sol–gel, mechanochemical, and current-assisted sintering processes; however, the majority of bulk ceramics are fabricated using a conventional solid-state reaction method, typically from mixed metal oxide–carbonate combinations. In this research work, a conventional solid-state reaction method was applied to fabricate the ceramic composition (NBT-BT-NBN$_{1-x}$T$_x$) in a series of (x = 0.00, 0.05, 0.10, 0.15, 0.20). The reactant materials were selected suitably with a preference for high purity and small particle sizes in the sub-micron range. The raw materials in powder forms were first heated at 100 °C overnight to remove the trapped water moisture contents. These reagents in powder form were weighed in stoichiometric ratios and ball-milled in plastic containers with $ZrO_2$ grinding balls, and ethanol was added as a lubricant for 24 h. The milling time is important in enhancing the dielectric properties of ceramic compositions. After finishing the ball milling, the resulted slurries were dried at 95 °C in an oven and then grinded to produce the powders. The milled powders of each batch sample were calcined at 850 °C for 2 h and again ball-milled for 12 h, after which they were dried to obtain fine powders. During the progress of the calcination process, the ball-milled powders of each batch sample were heated at a high temperature below their melting temperature, which enabled them to react for the development of the required crystalline phase. The calcination temperature varies from one material to another; therefore, significant attention was given to the calcination temperature of each batch sample. In this work, volatilization of the alkaline elements from the pellets was controlled by placing them in completely sealed alumina crucibles. A few drops of poly-vinyl alcohol (PVA) were added as a binder in the calcined samples and again grinded for 10 min, keeping each sample in a steel die with a 12 mm diameter, followed by placing it in an isotactic pressing machine and applying a uniaxial pressure of 150 MPa, in order to produce pellets of 1 mm thickness for each batch sample. All these pellets were initially heated at a temperature of 600 °C for 2 h to remove the PVA contents, and finally sintered in the temperature range of 1050–1150 °C for 2 h. All the green body

pellets were subjected to thermal treatment (sintering) below their melting temperature to enhance their mechanical strength for bonding their particles together. In this process, all the material samples were heated in the high-temperature range of 1050–1150 °C below their melting points. The best sintering temperature and most suitable stay times were used for the proper densification of each batch sample. For the optimization of the sintering temperature, each sample was subjected to various sintering temperatures, and then the sintered samples with the highest density were selected for the advanced characterization.

*3.3. Characterization Techniques*

At maximal density, pellets of each ceramic composition were first crushed to form powders for their phase evolution analysis with the advanced powder X-ray diffraction (PXRD) technique. The data were recorded at an angle of 2θ (degree) in the range of 10–70° at a scan rate of 0.02°/min using an X-ray diffractometer. Dense pellets of each sample were smoothly polished and etched thermally; then, they were coated in gold to avoid the charging effect during contact with an electron beam. The surface morphology of each sample was examined using a scanning electron microscope.

Furthermore, 0.70 mm thick ceramic samples were prepared and sintered at 1150 °C. Both faces of each sample were thoroughly polished before coating them in silver and heated at 800 °C for 2 h for the evaluation of the dielectric constant and dielectric loss for each batch sample, at frequencies of 1 kHz, 10 kHz, 100 kHz, and 1 MHz, against the temperature by recording at a scan rate of 3 °C/min with an LCR meter (Agilent E498, Santa Clara, CA, USA) in the temperature range of 20–500 °C. The thickness of each dense sample was decreased to 0.25 mm; then, the samples were coated with silver paste and heated at 800 °C for 2 h, and the polarization vs. electric field (P–E) hysteresis loops were recorded at 10 Hz for each batch sample of the fabricated ceramic compositions at the breakdown voltage using a ferroelectric analyzer (aixACCT TF-2000, UCSU, Germany).

## 4. Conclusions

A series (x = 0.00, 0.05, 0.10, 0.15, 0.20) of ceramic compositions of [0.80(0.92$Bi_{0.5}Na_{0.5}TiO_3$-0.08$BaTiO_3$)-0.20 $Na_{0.73}Bi_{0.09}NbO_3$-x$Ta_2O_5$], (NBT-BT-$NBN_{1-x}T_x$), were prepared using a conventional solid-state reaction method with doping of Ta. The effects of Ta doping on the as-fabricated ceramic compositions' phase evolution, microstructure development, electrochemical properties, and energy storage density properties were investigated using various advanced characterization techniques such as advanced powder X-ray diffractometry, scanning electron microscopy, and an LCR meter. A single perovskite phase developed in all the synthesized compositions; however, excessive addition of Ta as a dopant resulted in the development of secondary phases, appearing as impurities at x ≥ 0.1. A decreasing trend was observed in the grain size for x ≤ 0.05, and a maximum polarization ($P_m$) of ~22 $\mu C/cm^2$ and recoverable energy of 1.57 $J/cm^3$ with an efficiency of 80% at an applied external electric field of 175 kV/cm were obtained. These results demonstrate the enhanced ferroelectric characteristics of the as-synthesized electro-ceramic perovskite compositions, which are suitable for high-energy storage density applications in electro-ceramic capacitors.

**Author Contributions:** A.K. and T.M.K.: conceptualization, methodology, software, validation; J.W.: resources; A.M. and H.B.: data curation, writing—draft preparation; S.Z.K.: reviewing and editing; X.W. and N.S.G.: visualization, investigation, supervision. All authors have read and agreed to the published version of the manuscript.

**Funding:** This work was funded by the Drug Discovery Research Center, Southwest Medical University, Luzhou, China, under Grant No.42-00040176 awarded to TMK. This research work was partially supported by National Science Foundation of China (Grant No.21861004 and 22165003).

**Data Availability Statement:** Data that support the findings of this study are included within the article.

**Acknowledgments:** This research work was fully supported by the Drug Discovery Research Center, Southwest Medical University, Luzhou, China, under Grant No. 42-00040176, which was awarded to Taj Malook Khan.

**Conflicts of Interest:** The authors declare no conflict of interest.

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
