# Peer review of "Fabrication of Lead-Free Bismuth Based Electroceramic Compositions for High-Energy Storage Density Applications in Electroceramic Capacitors"

_catalysts, doi:10.3390/catal13040779_

Round 1

Reviewer 1 Report (New Reviewer)

The authors synthesized a series of Lead-free Bismuth based electro-ceramic perovskites with a conventional solid state reactions method. Meanwhile, the doping of Ta was achieved during the fabrication of compounds. Furthermore, the effect of Ta-doping on the performance was also systematically investigated. The structure was organized well. Nevertheless, there are still some following issues need to be further addressed be fore its acceptance. 

(1) The standard PDF card and pattern should be provided to demonstrate the phase purity and structure.

(2) TEM and HRTEM results should be given for the best sample.

(3) More discussions and comparisons on performance should be carried out for your samples.

(3) The language of this manuscript should be improved significantly. 

Author Response

Authors responses to reviewer comments

Q.1.The standard PDF card and pattern should be provided to demonstrate the phase purity and structure.

Ans. To address the comment of the honorable reviewer, the single perovskite phase Bi0.5N0.5TiO3 has the standard PDF card No.070-4780 and the secondary phase Bi2Ti2O7 has the standard PDF card No.32-0118. In contrast, the x-ray patterns have been provided for the phase purity of the as-fabricated ceramic composition in Figure 1 of the manuscript

Q.2 TEM and HRTEM results should be given for the best sample.

Answer 2. To address the comment of the honorable reviewer, the TEM and HRTEM facilities are not available in out institute. There are also some technical and financial issues to do these analysis from some other organization. It needs a lot of time to do TEM and HRTEM analysis, hence, it is hardly possible in the present circumstances

Q.3 More discussions and comparisons on performance should be carried out for your samples.

Ans. To address the comment of the honorable reviewer, the manuscript has revised well and then comparative analysis of this work is provided in Table 1 of the manuscript.

Q.4.The language of this manuscript should be improved significantly. 

Ans. To address the comment of the honorable reviewer, the manuscript has been revised several times for improving its English language.

Reviewer 2 Report (New Reviewer)

The authors introduced the dielectric and energy storage properties of the lead-free bismuth based electro ceramic composites. However, the figure quality is low, and the analysis is not persuasive.  I suggest the authors revise the manuscript based on my comments before further consideration for publication on Catalysts.

1. Figure 1 should include the material phase for different symbols.

2. The secondary phase in Figure 2 is too small to be observed. The authors might want to use energy dispersion spectroscopy (EDS) to verify that it is the secondary phase instead of a primary phase not grown large.

3. Caption for Figure 3(f) should be included. The caption for Figure 3(f) is too general and the content of Figure 3(f) is not included.

4. (a) and (b) are needed for Figure 4. The fonts for the left figure and the right figure are different. The authors might want to make it uniform. In addition, why only tanδ spectra with x=0 and x=0.05 are presented? In order to have comprehensive understanding, tanδ spectra with all five samples should be included.

5. (A)-(D) should be revised to (a)-(d) to make it uniform for the whole manuscript. More than one set of data is needed for Figure 5(b)-5(d) and error bars are needed to verify the results and conclusions.

Author Response

Responses to reviewer comments

Q.1. Figure 1 should include the material phase for different symbols.

Ans: To address the comment of the honorable reviewer, different symbols have been added in Figure 1 of the manuscript for material phases such as symbol  # " for the single perovskite phase of Bi0.5N0.5TiO3 and symbol "*" for the secondary phase Bi2Ti2O7 also known as bismuth pyrochlore. These symbols are also added in the manuscripts text (highlighted).

Q.2.The secondary phase in Figure 2 is too small to be observed. The authors might want to use energy dispersion spectroscopy (EDS) to verify that it is the secondary phase instead of a primary phase not grown large.

Ans: To address the comment of the honorable reviewer, the EDS facility is not available in our institute. There are also some technical and financial issues to get analyze from other organization. A lot of time needs to analyze the materials for EDS, hence, it is hardly possible in the present circumstances.

Q.3. Caption for Figure 3(f) should be included. The caption for Figure 3(f) is too general and the content of Figure 3(f) is not included.

Ans: To address the comment of the honorable reviewer, caption for 3(f) has been added in the manuscript and also detailed as highlighted in manuscript.

Q.4. (a) and (b) are needed for Figure 4. The fonts for the left figure and the right figure are different. The authors might want to make it uniform. In addition, why only tanδ spectra with x=0 and x=0.05 are presented? In order to have comprehensive understanding, tanδ spectra with all five samples should be included.

Ans: To address the comment of the honorable reviewer, all the spectra of five sample for tanδ (4a-e) have been added with uniform captions and discussed as highlighted in the manuscript

Q.5. (A)-(D) should be revised to (a)-(d) to make it uniform for the whole manuscript. More than one set of data is needed for Figure 5(b)-5(d) and error bars are needed to verify the results and conclusions.

Ans: To address the comment of the honorable reviewer, all errors indicated by reviewer have been removed for the uniformness of data throughout the manuscript.as highlighted.

Round 2

Reviewer 1 Report (New Reviewer)

The authors have addressed most concerns and the manuscript has been improved to a large extent. Thus, I think the manuscript can be accepted now. 

Author Response

We are very thankful to you for recommending our manuscript for the acceptance.

Reviewer 2 Report (New Reviewer)

The manuscript was revised based on my comments except the one that I asked for EDS measurement to prove the second phase. The second phase is the core to understand the dielectric responses according to the discussion in the manuscript. I insist to incorporate EDS measurements, otherwise the conclusions are all based on the authors' guess instead of scientific proof. 

Author Response

Responses to Reviewer

Dear respected editor,

            We appreciate your kind evaluation of our manuscript entitled “Fabrication of Lead-free Bismuth based Electro-ceramic Compositions for High Energy Storage Density Applications in Electro-ceramic Capacitors” in Journal of Catalysts.

Responses to reviewer comments

CommentsThe manuscript was revised based on my comments except the one that I asked for EDS measurement to prove the second phase. The second phase is the core to understand the dielectric responses according to the discussion in the manuscript. I insist to incorporate EDS measurements, otherwise the conclusions are all based on the authors' guess instead of scientific proof. 

Ans: To address the comment of the honorable reviewer, energy dispersive spectroscopic (EDS) measurements and spectrum for bismuth pyrochlore Bi2Ti2O7 as secondary phases have been added in manuscript that is depicted in Figure 2(e) while the secondary phases of Bi2Ti2O7 are also indicated with arrows in the SEM microstructures.

This manuscript is a resubmission of an earlier submission. The following is a list of the peer review reports and author responses from that submission.

Round 1

Reviewer 1 Report

Authors reported the Fabrication of Lead-free Tantalum doped Bismuth Based Electroceramic Compositions for High Energy Storage Density Applications in Electroceramic Capacitors. This manuscript is well written, and some results are interesting, however following revisions should be made before publication.

1.       How did authors determine the formula of electroceramics perovskite 0.80(0.92Bi1/5Na1/5TiO3-0.08BaTiO3)-0.20(Na0.73Bi0.09NbO3-xTa2O5)?

2.       The resolution of all figures is very poor.

3.       Please provide the SEM elemental mapping for NBT-BT-NBN1-xTx.

4.       In introduction section rational use of metal oxide as High Energy Storage Density Applications can be explained on the basis of following articles:

doi.org/10.1016/j.est.2023.106713, and https://doi.org/10.3390/catal10050546

5.       In the P-E hysteresis loops analysis Dielectric properties measurement background colour should be removed.

6.       There are so many typographical errors that should be corrected. For example…3.3.  .  Figure 1. Or figure 1.

Reviewer 2 Report

Reviewer's comments

The manuscript reports, Fabrication of Lead-free Bismuth-based Electro-ceramic compositions for high energy storage density applications in Electro-ceramic capacitors. The manuscript is primarily focused on the investigation of phases, microstructures, dielectrics and energy storage density applications of the electro-ceramic compositions by a solid-state reaction method. The obtained results are very interesting. I believe the manuscript could be considered for publication after a minor revision. Authors are suggested to address the following concerns.

1.     Introduction seems lengthy while discussion is short. Authors are suggested to shorten the introduction and provide more detailed discussion in the results and discussion aprt.

2.     Authors should explain the following parameters. How the average grain size for each composition was determined?

3.     How was the maximum density of each pellet determined? Although the SEM images of all are look equal dense.

4.     Secondary phases were observed in the PXRD patterns for x 0.01 contents. May the authors figure out a way to try to remove it?

5.     Authors are suggested to revise the conclusion, make it more coherent and concise.

6.     Figure titles and labels are not in a consistent and uniform format, such as font size and fonts style are different in every figure. Authors are suggested to use similar fonts size and format.

7.     How to differentiate between paraelectric ceramic and ferroelectric ceramics?  

8.     There are numerous typos and grammatical errors in manuscript. Please carefully revise the manuscript.